

# Construction and validation of a predictive model for the risk of malnutrition in hospitalized patients over 65 years of age with malignant tumours: a single-centre retrospective cross-sectional study

Xuexing Wang[1,*], Jie Chu[2,*], Chunmei Wei[1], Jinsong Xu[1], Yuan He[3] and Chunmei Chen[4]

[1] Department of Oncology, Anning First People's Hospital Affiliated to Kunming University of Science and Technology, Kunming, Yunnan, China
[2] Department of Oncology, West China-Ziyang Hospital, Sichuan University, Ziyang, Sichuan, China
[3] Department of Geriatric Oncology, The Third Affiliated Hospital of Kunming Medical University, Kunming, Yunnan, China
[4] Department of Pharmacy, Anning First People's Hospital Affiliated to Kunming University of Science and Technology, Kunming, Yunnan, China
* These authors contributed equally to this work.

Corresponding author
Xuexing Wang, 906013302@qq.com

## ABSTRACT

**Background:** Nutritional status is a critical indicator of overall health in individuals suffering from malignant tumours, reflecting the complex interplay of various contributing factors. This research focused on identifying and analysing the factors influencing malnutrition among older patients aged ≥65 with malignant tumours and aimed to develop a comprehensive risk model for predicting malnutrition.

**Methods:** This study conducted a retrospective analysis of clinical data from 3,387 older inpatients aged ≥65 years with malignant tumours collected at our hospital from July 1, 2021, to December 31, 2023. The dataset was subsequently divided into training and validation sets at an 8:2 ratio. The nutritional status of these patients was evaluated using the Nutritional Risk Screening Tool 2002 (NRS-2002) and the 2018 Global Leadership Initiative on Malnutrition (GLIM) Standards for Clinical Nutrition and Metabolism. Based on these assessments, patients were categorized into either malnutrition or non-malnutrition groups. Subsequently, a risk prediction model was developed and presented through a nomogram for practical application.

**Results:** The analysis encompassed 2,715 individuals in the development cohort and 672 in the validation cohort, with a malnutrition prevalence of 40.42%. A significant positive correlation between the incidence of malnutrition and age was observed. Independent risk factors identified included systemic factors, tumour staging (TNM stage), age, Karnofsky Performance Status (KPS) score, history of alcohol consumption, co-infections, presence of ascites or pleural effusion, haemoglobin (HGB) levels, creatinine (Cr), and the neutrophil-to-lymphocyte ratio (NLR). The predictive model exhibited areas under the curve (AUC) of 0.793 (95% confidence interval (CI) [0.776–0.810]) for the development cohort and 0.832 (95% CI [0.801–0.863]) for the validation cohort. Calibration curves indicated Brier scores of 0.186 and 0.190, while the Hosmer–Lemeshow test yielded chi-square values of 5.633

and 2.875, respectively ($P > 0.05$). Decision curve analysis (DCA) demonstrated the model's clinical applicability and superiority over the NRS-2002, highlighting its potential for valuable clinical application.

**Conclusion:** This study successfully devised a straightforward and efficient prediction model for malnutrition among older patients aged 65 and above with malignant tumours. The model represents a significant advancement as a clinical tool for identifying individuals at high risk of malnutrition, enabling early intervention with targeted nutritional support and improving patient outcomes.

## BACKGROUND

Disease-related malnutrition (DRM) represents a discrepancy between the intake of nutrients and energy and their respective requirements, resulting in metabolic and functional alterations in nutritional status and body composition (*García-García et al., 2023*). Malnutrition is a significant health and well-being issue, particularly in developing countries. Currently, malnutrition among hospitalized patients has garnered widespread attention in public health and epidemiology across the globe (*Rahman et al., 2021*). Patients in China, especially those of older age and those with malignancies, face a high risk of malnutrition. Up to 20% of patients with cancer may succumb to the harmful effects of malnutrition rather than the malignancy itself (*García-García et al., 2023*; *Yan et al., 2022*). Research consistently indicates a strong association between nutritional status and various outcomes in patients with cancer, including quality of life, treatment efficacy, and overall prognosis (*Zhang et al., 2022, 2023*). Malnutrition has been identified as an independent risk factor affecting the prognosis of patients with malignant tumours (*Zhou et al., 2023*). Factors such as advanced age, cognitive decline, comorbidities, excessive polypharmacy, depression, loss of appetite, prolonged hospitalization, and immune dysfunction are primarily associated with malnutrition in these patients (*Kaya et al., 2019*). Although several validated tools (*Basile et al., 2019*; *Alston et al., 2022*; *Zhou et al., 2022*), such as the Malnutrition Screening Tool (MST) and the Patient-Generated Subjective Global Assessment (PG-SGA), are widely used in diagnosing malnutrition, particularly in older adults and cancer patients, these tools may still not fully address the specific nutritional challenges faced by elderly patients with malignant tumors. As such, universally accepted diagnostic criteria for malnutrition in this population are still evolving. Various international authorities continue to revise and refine these criteria. The 2018 Global Leadership Initiative on Malnutrition (GLIM) Standards were introduced to create a globally consistent framework for malnutrition assessment in adult inpatients. This framework emphasizes a two-phased approach to malnutrition assessment into two steps: initial "nutritional screening" followed by a detailed "diagnostic assessment" (*Zhou et al., 2022*; *Takimoto et al., 2022*). In China, medical professionals frequently evaluate patients' clinical and nutritional statuses using parameters (*Kaya et al., 2019*; *Wang et al., 2023*) such as daily dietary monitoring, anthropometric measurements such as body mass

index (BMI), weight, calf circumference, and skinfold thickness, and laboratory blood tests, such as serum albumin and total cholesterol levels. Nutritional assessment tools, including the Mini Nutritional Assessment (MNA), the Geriatric Nutritional Risk Index (GNRI), the Nutritional Risk Screening 2002 (NRS-2002), the Generalized Screening Tool for Malnutrition (GSTM), and the Subjective Global Assessment (SGA), have demonstrated reliability across various studies for evaluating malnutrition. However, these tools are not tailored for older patients with malignant tumours. Moreover, previous research frequently relied on traditional methods such as logistic regression to identify malnutrition risk factors without considering the specific pathogenesis related to older cancer patients. Therefore, pinpointing critical factors influencing malnutrition through relevant indicators in this demographic and establishing a multidimensional risk prediction model for the early identification of at-risk individuals presents a more effective strategy for preventing malnutrition.

## METHODS

### Study design and participants

This retrospective cross-sectional study was conducted at the Anning First People's Hospital Affiliated to Kunming University of Science and Technology. This study retrospectively analyzed clinical data from 3,387 older inpatients aged 65 years and older with malignant tumours who were admitted between July 1, 2021, and December 31, 2023. The data were divided into training and validation sets based on patient admission timeframes, maintaining an 8:2 ratio, with the training set comprising 2,715 patients admitted between July 1, 2021, and July 31, 2023, and the validation set consisting of 672 patients admitted from August 1, 2023, to December 31, 2023.This study was conducted following the Declaration of Helsinki by the World Medical Association. It was approved by the Ethical Review Committee of the First People's Hospital of Anning, affiliated with Kunming University of Science and Technology (Ethical Approval No. 2024-020 (Self-horizontal)-01). Given its retrospective nature, the requirement for informed consent was waived. All data were treated confidentially and analyzed anonymously to protect patient privacy, no information identifying individual participants was available to the authors during or after data collection.

The inclusion criteria for this study were as follows: (I) patients aged 65 years or older; (II) patients with histopathologically confirmed malignant tumours; and (III) patients with complete data available during their hospitalization, retrievable from the Hospital Information System (HIS), Laboratory Information Management System (LIS), or Electronic Medical Record System (EMRS), encompassing comprehensive information. The exclusion criteria encompassed patients with severe cardiac, hepatic, renal, or pulmonary dysfunction, and those with bleeding from vital organs such as the digestive, respiratory tract, or liver. Patients who were admitted to the Intensive Care Unit (ICU) or Coronary Care Unit (CCU), those anticipated to have a life expectancy of <3 months and those with >20% missing data were also excluded from the study.

## Sample size estimation

The calculation of an adequate sample size for predictive studies, encompassing both modelling and validation phases, depends on the number of outcome events. Following guidelines from predictive modelling research, including those described by *Hosmer, Lemeshow & Sturdivant (2013)*, the number of positive malnutrition cases should ideally exceed the number of examined risk factors by 5–10 times to mitigate the likelihood of overfitting and prediction errors. In this study, 26 risk factors were considered. A minimum of 130 malnutrition cases was deemed necessary to maintain prediction accuracy. The modelling sample size was 3,387, with 1,369 positive instances, which exceeds the requisite minimum to ensure a robust assessment.

## Data extraction and nutritional status evaluation

The analysis incorporated a range of variables, including demographic details, medical history, clinical observations, and laboratory indicators from patients admitted to our hospital with malignant tumours between July 2021 and December 2023. The selection of risk factor variables for initial screening was informed by potential causes of malnutrition in older patients with malignant tumours, existing literature, and insights derived from clinical experience and consultations with nutrition experts. The collected demographic information encompassed gender, age, marital status, Hukou (household registration) status, and BMI. Moreover, medical history was documented, encompassing habits such as smoking and alcohol consumption, and the presence of conditions such as hypertension, type 2 diabetes mellitus, coronary artery disease, and hyperlipidaemia. Details of any surgical interventions within the last year, and the incidences of co-infections were recorded. The clinical information gathered encompassed the clinical diagnosis, the TNM stage of the tumour, treatment plans, the presence of thoracic and abdominal effusions, the NRS-2002 score, and the KPS (Karnofsky Performance Status) score.

Laboratory findings considered in this analysis encompassed red blood cell (RBC) count, white blood cell (WBC) count, haemoglobin (HGB) levels, platelet (PLT) count, neutrophil-to-lymphocyte ratio (NLR), serum albumin (ALB), aspartate aminotransferase (AST),alanine aminotransferase (ALT), total bilirubin (TBIL), creatinine (Cr), and blood urea nitrogen (BUN) levels.

The GLIM criteria were adopted for malnutrition diagnosis. Nutritional screening constituted the first phase, using the NRS-2002 as the initial tool. A score of ≥3 on the NRS-2002, verified through medical records, indicated a malnutrition risk. Subsequently, the second stage involved a detailed malnutrition assessment based on five criteria: involuntary weight loss (>5% within 6 months or >10% at 6 months), low BMI (Asian standard (*Kanazawa et al., 2005*): BMI <18.5 kg/m$^2$ (<70 years), <20 kg/m$^2$ (≥70 years)), and decreased muscle mass (below the standard values for different body composition measures; Decreased muscle mass is defined as a reduction in muscle mass below accepted reference values, determined through physical measurements conducted by clinical staff using standardized anthropometric techniques. These measurements, including skinfold thickness and circumferences, were used to assess and compare muscle mass to reference values, identifying significant reductions where applicable as part of the phenotypic

criteria. The remaining two etiologic criteria involve reduced food intake or absorption (food intake <50% for >1 week, or reduced intake for >2 weeks, or chronic gastrointestinal dysfunction impairing food digestion and absorption), and the disease/inflammation burden. Inflammation burden was assessed using clinical biomarkers with established laboratory reference ranges. Specifically, inflammation was quantified using C-reactive protein (CRP) levels and erythrocyte sedimentation rate (ESR). For this study, Significant inflammation was defined according to the measurement reference ranges provided by our testing instruments, with CRP levels >10 mg/L and/or ESR >30 mm/hr as indicators of elevated inflammation. To diagnose malnutrition, at least one phenotypic criterion and one etiologic criterion, including these inflammation cut-off values, must be met.

## Model development and validation process

In this study, 2,715 patients admitted consecutively between July 1, 2021, and July 31, 2023, were included in the training set. Meanwhile, 672 patients admitted from August 1, 2021, to December 31, 2023, comprised the validation set, maintaining an 8:2 ratio. The modeling process classified malnutrition cases in the training set as the dependent variable, thereby differentiating participants into malnutrition and non-malnutrition groups. A forward selection method was employed to identify and select statistically significant variables ($P < 0.05$) for inclusion in the multivariate logistic regression analysis. This analysis was conducted to evaluate the influence of these factors on the likelihood of malnutrition, with significance assessed using *P-values* and the impact quantified using odds ratios. Column plots were generated using the "rms" package in R software (version 4.2.0) (*R Core Team, 2023*; *Harrell, 2023*). The model's validity was confirmed by employing Statistics software (version 26.0) (IBM, Armonk, NY, USA) to generate receiver operating characteristic (ROC) curves, calculate the area under the curve (AUC) for both sets and compare the model's predictive accuracy against the NRS-2002 scale, thereby evaluating its discriminative capabilities. The calibration curves for the training and validation sets were developed using the "rms" package in R programming language (version R 4.2.0). Model calibration was assessed through visual inspection and the Hosmer-Lemeshow test, a statistical test used to evaluate the goodness-of-fit for logistic regression models by comparing the observed and predicted event rates across different subgroups (*Hosmer, Lemeshow & Sturdivant, 2013*). The test helps to determine whether the model adequately fits the data or if there are systematic discrepancies between the observed and predicted values. Clinical applicability was evaluated through decision curve analysis (DCA) using the "rmda" package. The model underwent internal validation with bootstrap resampling conducted 1,000 times, considering $P < 0.05$ statistically significant.

## Statistical analysis

Data were inputted using Excel version 2022, with a double-entry method applied to minimize data entry errors. Data summarization and analyses were performed using SPSS 26.0 software (IBM SPSS 26.0; SPSS Inc., Chicago, Illinois, USA). Measurement data following a normal distribution were presented as x̄ ± s, and t-tests were applied for comparisons between groups. Count or categorical data were expressed as frequencies (n)

and percentages (%), with chi-square tests or Fisher's exact test used for group comparisons. Medians and interquartile ranges represented measurement data not adhering to a normal distribution (M (P25, P75)). Regarding missing data, particularly for muscle mass measurements, we utilized multiple imputation methods. This approach involves generating several complete datasets to replace missing values, performing analyses on each dataset, and then combining the results to provide a comprehensive estimate. This method reduces bias and enhances the robustness of our findings despite the missing data. The nomogram's performance was determined using the concordance index (C-index), which evaluates the agreement between predicted probabilities and actual observations of malnutrition. Bootstrapping with 500 resamples was performed. A higher C-index signifies more accurate prognostic differentiation. The total points for each patient in the validation cohort were calculated utilizing the constructed nomogram to assess the risk of malnutrition. The C-index, calibration curve, and DCA were derived from this analysis using the R package (version 4.2.0) for all nomogram-related computations.

## RESULTS

### Characteristics and predictive factors of malnutrition in development and validation cohorts

After applying the criteria, 3,387 patients were included in the study, as shown in Fig. 1. The characteristics of patients aged ≥65 years in the development and validation cohorts are listed in Table 1. These cohorts displayed similarities in gender, Hukou status, TNM staging, and Karnofsky Performance Status (KPS) scores. However, differences were observed in alcohol consumption history, hypertension, type 2 diabetes mellitus, hyperlipidaemia, and coronary heart disease (CHD). Table S1 presents an exhaustive analysis of malnutrition indicators in patients aged 65 and older diagnosed with malignancies. The table encompasses detailed data on phenotypic factors, including the number of patients experiencing significant weight loss, the prevalence of low body mass index (BMI), and the degree of muscle mass reduction. Specifically, Table S1 delineates the proportion of patients experiencing weight loss, stratified by severity, alongside the count of patients exhibiting low BMI values below predefined thresholds. Additionally, it elucidates the methodologies employed by clinical staff to assess muscle mass, encompassing the anthropometric techniques utilized, and details the approach to handling missing muscle mass data through multiple imputation methods. Table 2 summarizes the association between clinical characteristics and malnutrition in the development cohort of older cancer inpatients. No significant differences were noted in gender, smoking history, hyperlipidaemia, and alanine aminotransferase (ALT) between the negative and positive malnutrition groups ($P > 0.05$). Conversely, factors such as age, marital status, Hukou status, tumour type, TNM staging, surgical history within the past 12 months, KPS score, alcohol consumption history, co-infection, hypertension, type 2 diabetes mellitus, CHD, presence of ascites or pleural effusion, counts of WBC, RBC, PLT, HGB, total bilirubin, aspartate aminotransferase (AST), Cr, urea, and the NLR were

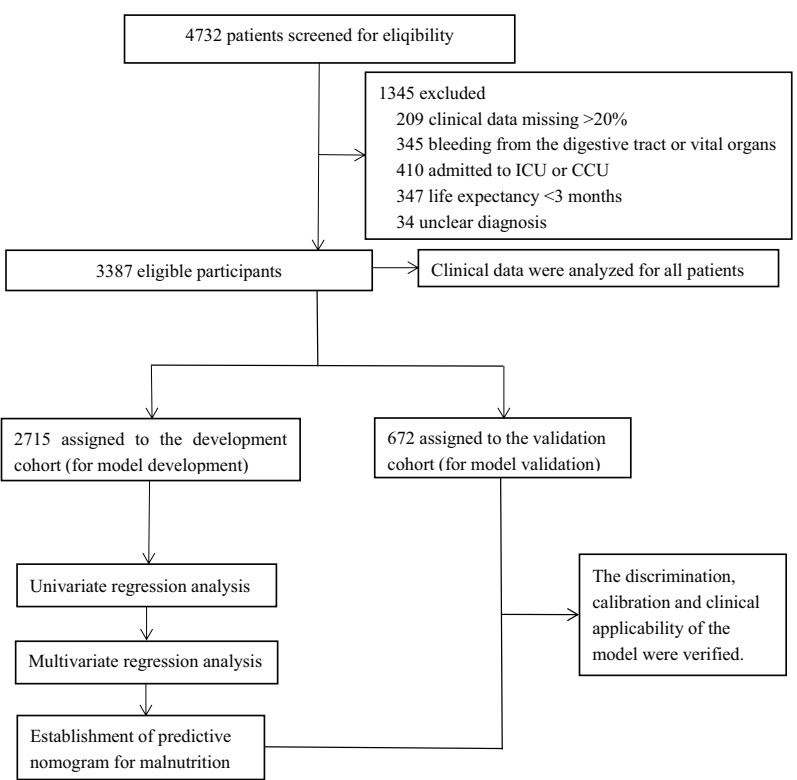

**Figure 1 Experimental roadmap for study.** This flow diagram indicates the workflow of the method present in this study.

significantly associated with the development of malnutrition. Table S2 provides the methods used for assigning variables in the logistic regression analysis during model construction. This includes strategies such as setting threshold values, integrating laboratory reference ranges, applying clinical expertise, and categorizing variables to identify the most effective assignment methods. Table S3 presents the Univariate Logistic Regression Analysis Identifying Risk Factors for Malnutrition in the Training Cohort, showcasing the initial screening of potential predictors. The multivariate analyses identified 10 variables as independent risk factors for malnutrition (Table 3). These included the system of the tumour, patient age, KPS score, TNM staging, history of alcohol consumption, co-infection presence, ascites or pleural effusion, HGB and Cr levels, and the NLR, all of which were significant predictors of malnutrition in older patients with cancer.

## Prevalence of malnutrition in older patients with malignant tumours

Analysis of the case data from the development and validation cohorts revealed 1,369 instances of malnutrition, resulting in a prevalence rate of 40.42%. Analysis of malnutrition rates by age, gender, Hukou status, and tumour type revealed that older patients were most susceptible to malnutrition, with men experiencing a higher prevalence than women. Patients from rural areas had a higher malnutrition rate than those from urban areas; digestive system tumours were most frequently associated with malnutrition.

| Characteristic | Entire cohort (n = 3,387) No. (%) | Development cohort (n = 2,715) No. (%) | Validation cohort (n = 672) No. (%) | P-value |
|---|---|---|---|---|
| Age; M (P25, P75) | 73 (68,78) | 73 (68,78) | 72 (69,77) | 0.194 |
| Gender | | | | 0.020 |
| Male | 2,120 (62.6) | 1,676 (61.7) | 444 (66.1) | |
| Female | 1,267 (37.4) | 1,039 (38.3) | 228 (33.9) | |
| Marital state | | | | 0.440 |
| Married | 2,756 (81.4) | 2,211 (81.4) | 545 (81.1) | |
| Other (Divorced/Widowed/Single) | 631 (18.6) | 504 (18.6) | 127 (18.9) | |
| Hukou | | | | <0.001 |
| Rural | 1,354 (40.0) | 1,007 (37.1) | 347 (51.6) | |
| Urban | 2,033 (60.0) | 1,708 (62.9) | 325 (48.4) | |
| System | | | | 0.170 |
| Respiratory system | 1,211 (35.8) | 985 (36.2) | 226 (33.6) | |
| Digestive system | 1,242 (36.7) | 988 (36.4) | 254 (37.8) | |
| Reproductive system | 475 (14.0) | 385 (14.2) | 90 (13.4) | |
| Blood system | 117 (3.5) | 86 (3.2) | 31 (4.6) | |
| Urinary system | 159 (34.7) | 133 (4.9) | 26 (3.9) | |
| Endocrine system, Motor system, Skin, and Appendicular ducts | 76 (2.2) | 59 (2.2) | 17 (2.5) | |
| Others | 107 (3.2) | 79 (2.9) | 28 (4.2) | |
| TNM staging | | | | 0.003 |
| Stage I–III | 1,893 (55.9) | 1,566 (57.7) | 348 (51.8) | |
| Stage IV | 1,473 (43.5) | 1,149 (42.3) | 324 (48.2) | |
| Surgical history (last 12 months) | | | | 0.072 |
| No | 2,600 (76.8) | 2,099 (77.3) | 501 (74.6) | |
| Yes | 787 (23.2) | 616 (22.7) | 171 (25.4) | |
| KPS score | | | | 0.006 |
| ≥90 points | 2,287 (67.5) | 1,861 (68.5) | 426 (63.4) | |
| ≤80 points | 1,100 (32.5) | 854 (31.5) | 246 (36.6) | |
| Smoking history | | | | 0.256 |
| No | 2,436 (71.9) | 1,960 (72.2) | 476 (70.8) | |
| Yes | 951 (28.1) | 755 (27.8) | 196 (29.2) | |
| Alcohol consumption history | | | | 0.002 |
| No | 2,872 (84.8) | 2,327 (85.7) | 545 (81.1) | |
| Yes | 515 (15.2) | 388 (14.3) | 127 (18.9) | |
| Co-infection | | | | |
| No | 2,873 (84.8) | 2,311 (85.1) | 562 (83.6) | 0.183 |
| Yes | 514 (15.2) | 404 (14.9) | 110 (16.4) | |
| Hypertension | | | | 0.030 |
| No | 1,996 (58.9) | 1,622 (59.7) | 374 (55.7) | |
| Yes | 1,391 (41.1) | 1,093 (40.3) | 298 (44.3) | |
| Type 2 diabetes mellitus | | | | |
| No | 2,763 (81.6) | 2,239 (82.5) | 524 (78.0) | 0.002 |
| Yes | 624 (18.2) | 476 (17.5) | 148 (22.0) | |

Table 1 Demographic and clinical characteristics of patients.

| Characteristic | Entire cohort (n = 3,387) No. (%) | Development cohort (n = 2,715) No. (%) | Validation cohort (n = 672) No. (%) | P-value |
|---|---|---|---|---|
| Hyperlipidaemia | | | | |
| No | 2,505 (74.0) | 1,976 (72.8) | 529 (78.7) | 0.001 |
| Yes | 882 (26.0) | 739 (27.2) | 143 (21.3) | |
| Coronary heart disease (CHD) | | | | |
| No | 2,937 (86.7) | 2,366 (87.1) | 571 (85.0) | 0.001 |
| Yes | 450 (13.3) | 349 (12.9) | 101 (15.0) | |
| Ascites or pleural effusion | | | | |
| No | 3,041 (89.8) | 2,439 (89.8) | 602 (89.6) | 0.447 |
| Yes | 346 (10.2) | 276 (10.2) | 70 (10.4) | |

**Table 2 Demographic and clinicopathological characteristics of patients in the training cohort.**

| Characteristic | Non-malnutrition (n = 1,634) | Malnutrition (n = 1,081) | $x^2$ | P value |
|---|---|---|---|---|
| Demographic characteristics | | | | |
| Age (years) | | | 130.382 | <0.001 |
| 65–70 | 715 (43.8) | 348 (32.2) | | |
| 71–75 | 472 (28.9) | 268 (24.8) | | |
| 76–80 | 282 (17.3) | 187 (17.3) | | |
| 81–85 | 133 (8.1) | 192 (17.8) | | |
| ≥86 | 32 (2.0) | 86 (8.0) | | |
| Gender | | | 0.121 | 0.379 |
| Male | 1,013 (62.0) | 663 (61.3) | | |
| Female | 621 (38.0) | 418 (38.7) | | |
| Marital state | | | 9.352 | 0.001 |
| Married | 1,361 (83.3) | 850 (78.6) | | |
| Other (Divorced/Widowed/Single) | 273 (16.7) | 231 (21.4) | | |
| Hukou | | | 50.025 | <0.001 |
| Rural | 477 (61.1) | 458 (38.9) | | |
| Urban | 1,157 (65.0) | 623 (35.0) | | |
| System | | | 10.481 | 0.001 |
| Digestive system | 606 (37.1) | 468 (43.3) | | |
| Others | 1,028 (62.9) | 613 (56.7) | | |
| TNM staging | | | 27.859 | <0.001 |
| Stage I–III | 1,009 (61.8) | 557 (61.9) | | |
| Stage IV | 625 (38.2) | 524 (38.1) | | |
| Surgical history (last 12 months) | | | 11.525 | <0.001 |
| No | 1,227 (75.1) | 872 (80.7) | | |
| Yes | 407 (24.9) | 209 (19.3) | | |

(Continued)

| Characteristic | Non-malnutrition ($n = 1,634$) | Malnutrition ($n = 1,081$) | $x^2$ | P value |
|---|---|---|---|---|
| KPS score | | | 357.617 | <0.001 |
| ≥90 points | 1,344 (82.3) | 517 (47.8) | | |
| ≤80 points | 290 (17.7) | 564 (52.2) | | |
| Smoking history | | | 1.586 | 0.112 |
| No | 1,194 (73.1) | 766 (70.9) | | |
| Yes | 440 (26.9) | 315 (29.1) | | |
| Alcohol consumption history | | | 3.021 | 0.047 |
| No | 1,416 (86.7) | 911 (84.3) | | |
| Yes | 218 (13.3) | 170 (15.7) | | |
| Co-infection | | | 19.733 | <0.001 |
| No | 1,431 (87.6) | 880 (81.4) | | |
| Yes | 203 (12.4) | 201 (18.6) | | |
| Hypertension | | | 64.148 | <0.001 |
| No | 876 (53.6) | 746 (69.0) | | |
| Yes | 758 (46.4) | 335 (31.0) | | |
| Type 2 diabetes mellitus | | | 6.925 | 0.005 |
| No | 1,322 (80.9) | 917 (84.8) | | |
| Yes | 312 (19.1) | 164 (15.2) | | |
| Hyperlipidaemia | | | 0.059 | 0.421 |
| No | 1,192 (72.9) | 784 (72.5) | | |
| Yes | 442 (27.1) | 297 (27.5) | | |
| Coronary heart disease (CHD) | | | 4.425 | 0.020 |
| No | 1,406 (86.0) | 960 (88.8) | | |
| Yes | 228 (14.0) | 121 (11.2) | | |
| Ascites or pleural effusion | | | 42.260 | <0.001 |
| No | 1,518 (92.9) | 921 (85.2) | | |
| Yes | 116 (7.1) | 160 (14.8) | | |
| Clinicopathological characteristics | | | | |
| WBC ($10^9$/L) | | | 23.111 | <0.001 |
| <4 | 314 (19.2) | 185 (17.1) | | |
| 4–10 | 1,148 (70.3) | 714 (66.0) | | |
| >10 | 172 (10.5) | 182 (16.8) | | |
| RBC ($10^9$/L) | | | 85.139 | <0.001 |
| <4 | 497 (30.4) | 518 (47.9) | | |
| ≥4 | 1,137 (69.6) | 563 (52.1) | | |
| PLT ($10^9$/L) | | | 22.985 | <0.001 |
| <100 | 111 (6.8) | 87 (8.0) | | |
| 100–300 | 1,380 (84.5) | 840 (77.7) | | |
| >300 | 143 (8.8) | 154 (14.2) | | |
| HGB (g/L) | | | 160.645 | <0.001 |
| ≤120 | 385 (23.6) | 507 (46.9) | | |

| Characteristic | Non-malnutrition ($n = 1,634$) | Malnutrition ($n = 1,081$) | $x^2$ | P value |
|---|---|---|---|---|
| >120 | 1,249 (76.4) | 574 (53.1) | | |
| Total bilirubin (µmol/L) | | | 10.488 | 0.001 |
| ≤26 | 1,533 (93.8) | 978 (90.5) | | |
| >26 | 101 (6.2) | 103 (9.5) | | |
| ALT (U/L) | | | 0.180 | 0.358 |
| ≤40 | 1,427 (87.3) | 950 (87.9) | | |
| >40 | 207 (12.7) | 131 (12.1) | | |
| AST (U/L) | | | 16.538 | <0.001 |
| ≤35 | 1,359 (83.2) | 831 (76.9) | | |
| >35 | 275 (16.8) | 250 (23.1) | | |
| Creatinine (Cr; µmol/L) | | | 7.485 | 0.003 |
| ≤81 | 999 (62.0) | 718 (67.2) | | |
| >81 | 611 (38.0) | 350 (32.8) | | |
| Urea (mmol/L) | | | 18.657 | <0.001 |
| ≤8.8 | 1,441 (88.8) | 892 (83.0) | | |
| >8.8 | 151 (11.2) | 183 (17.0) | | |
| NLR | | | 100.169 | <0.001 |
| ≤3.24 | 1,049 (64.5) | 484 (45.0) | | |
| >3.24 | 577 (35.5) | 591 (55.0) | | |

**Table 3 Multivariable logistic regression analysis identifying malnutrition risk factors in the training cohort.**

| Characteristics | OR (95% CI) | P |
|---|---|---|
| System | | |
| Others | Reference | |
| Digestive system | 1.216 [1.016–1.457] | 0.033 |
| Age (years) | | |
| 65–70 | Reference | |
| 71–75 | 1.273 [1.021–1.588] | 0.032 |
| 76–80 | 1.303 [1.009–1.683] | 0.043 |
| 81–85 | 2.356 [1.763–3.149] | <0.001 |
| ≥86 | 4.141 [2.557–6.706] | <0.001 |
| KPS score | | |
| ≥90 points | Reference | |
| ≤80 points | 4.619 [3.834–5.566] | <0.001 |
| TNM staging | | |
| Stage I–III | Reference | |
| Stage IV | 1.081 [1.033–1.131] | 0.001 |
| Alcohol consumption history | | |
| No | Reference | |
| Yes | 3.865 [1.087–13.744] | 0.037 |

(Continued)

| Table 3 (continued) | | |
|---|---|---|
| Characteristics | OR (95% CI) | P |
| Co-infection | | |
| No | Reference | |
| Yes | 17.54 [3.546 to −83.333] | <0.001 |
| Ascites or pleural effusion | | |
| No | Reference | |
| Yes | 5.765 [1.205–25.578] | 0.028 |
| HGB (g/L) | | |
| ≤120 | Reference | |
| >120 | 0.441 [0.365–0.531] | <0.001 |
| Creatinine (Cr; μmol/L) | | |
| ≤81 | Reference | |
| >81 | 2.268 [1.883–2.740] | <0.001 |
| NLR | | |
| ≤3.24 | Reference | |
| >3.24 | 1.677 [1.394–2.016] | <0.001 |

Stage IV tumours were most frequently associated with malnutrition, with the distribution of malnutrition rates across various subgroups depicted in Fig. 2.

## Development of the malnutrition prediction nomogram

The prediction model's 10 independent variables (age, presence of a digestive system tumour, TNM staging, KPS score, history of alcohol consumption, history of infections, presence of combined thoracic and abdominal fluid, HGB levels, Cr levels, and the NLR) were employed to construct a nomogram. The logistic regression analysis identified these variables as significant predictors of malnutrition. Weights for each predictor in the nomogram were derived from their respective β-coefficients. The instructions for using the nomogram are outlined in the caption of Fig. 3A, which includes a comprehensive guide. Figure 3B displays an interactive version of the nomogram, enhancing user interaction and interpretation.

To use the nomogram, a vertical line was drawn from the point on the horizontal axis corresponding to each independent variable's value for a patient to the "Point" value representing a specific score. The scores for the five independent variables were summed to calculate the total score, after which a vertical line was drawn downwards. The predicted risk of malnutrition for the patient was determined by locating the corresponding point on the horizontal axis labelled "Risk".

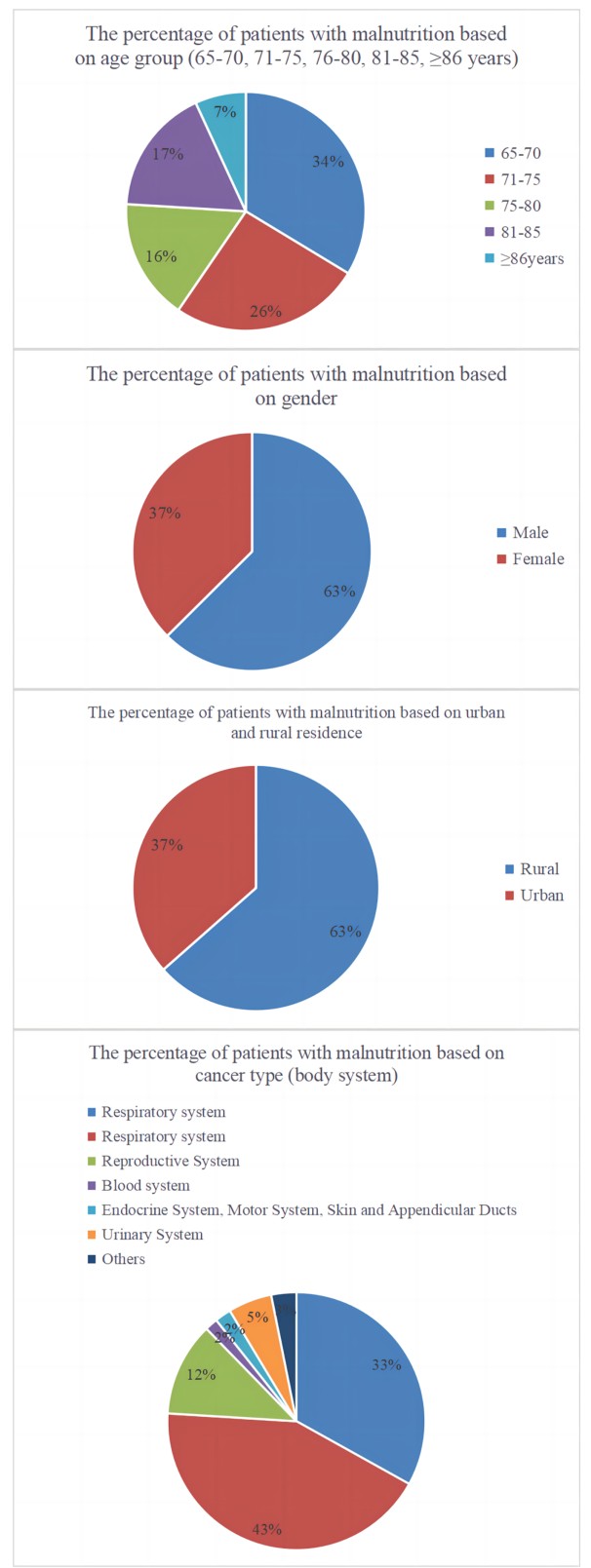

**Figure 2 Pie charts representing the distribution of malnutrition according to age, gender, residency status, and tumour type.**

A

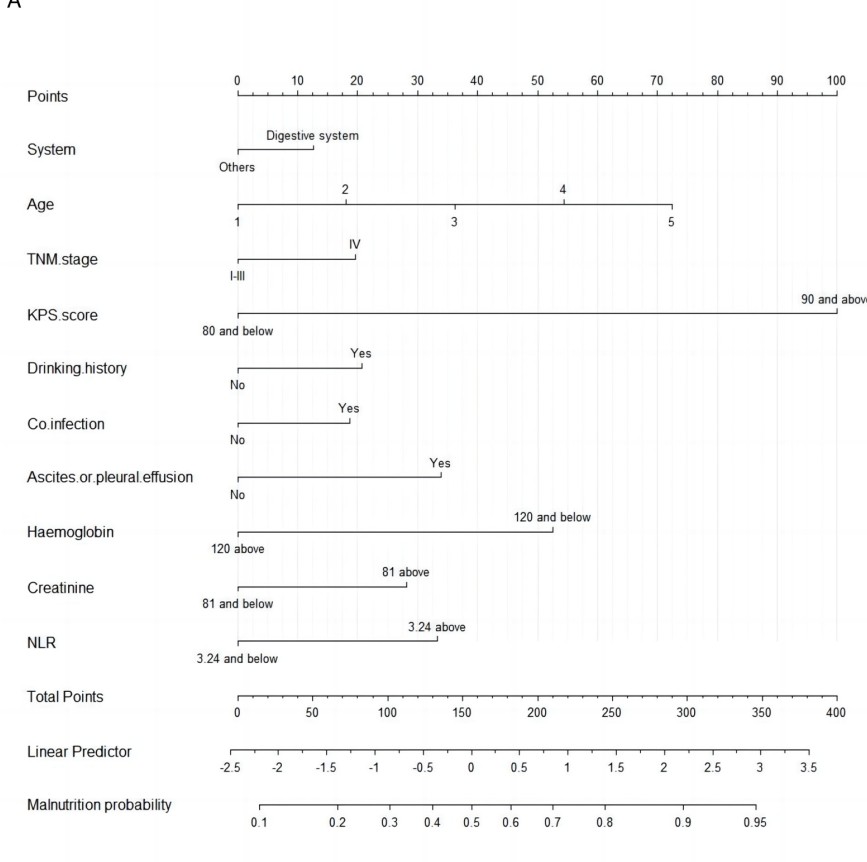

B

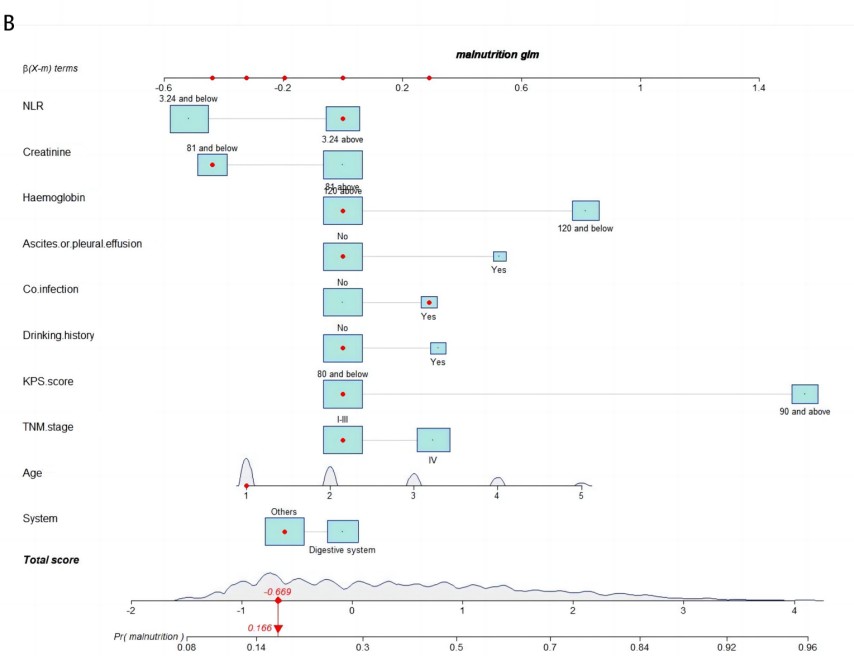

**Figure 3 Nomogram A and interactive nomogram B for malnutrition prediction in patients aged ≥65 years with tumours.** Points for each f (Age categorization: 1 = 65–70, 2 = 71–75, 3 = 76–80,

**Figure 3 (continued)**
4 = 81–85, 5 => 85 years). There are 10 factors in the malnutrition prediction nomogram. Points for each factor are obtained from the top scale and summed to generate a total score. This total score is projected onto the lower scale to estimate an individual's malnutrition risk. For example, the patient's TNM stage is located, and a line is drawn straight upward to the "Points" axis to determine the score associated with that TNM stage. The process is repeated for each variable, the scores for each covariate are summed, and this sum is located on the "Total Points" axis. A line is drawn straight down to determine the probability of malnutrition.                                                               

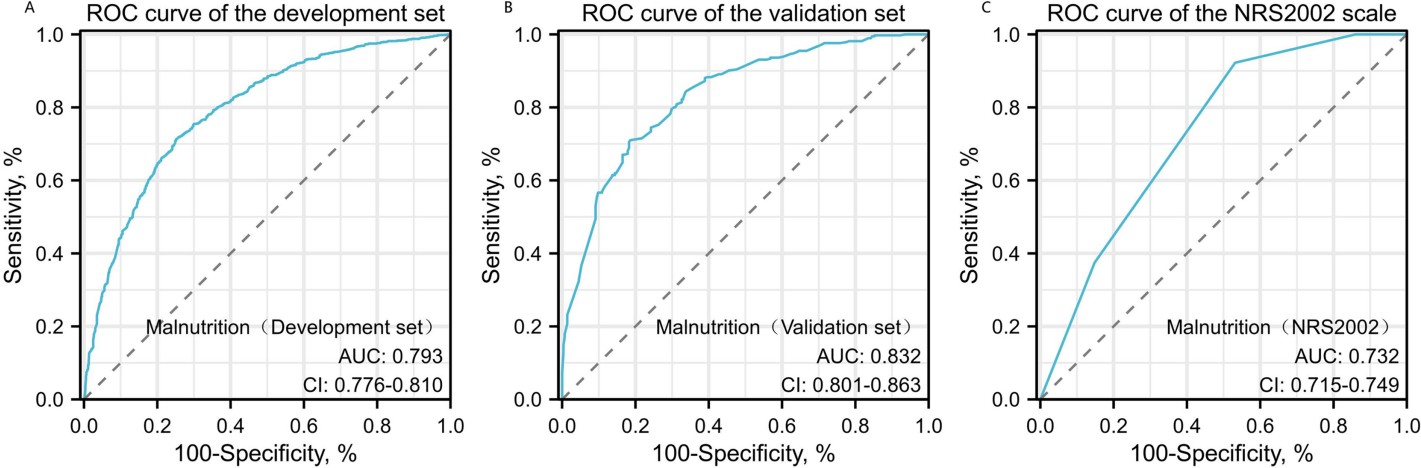

**Figure 4 Receiver operating characteristic (ROC) curve analysis for malnutrition prediction.** ROC curves for malnutrition prediction in the training set (A) and the testing set (B) are compared against the NRS2002 scale (C). *P*-values are two-sided. AUC values with 95% CI for the training set, testing set, and NRS2002 scale are 0.793 (95% CI [0.776–0.810]), 0.832 (95% CI [0.801–0.863]), and 0.737 (95% CI [0.715–0.749]), respectively, with Delongtest *P* > 0.05.                                                         

## Evaluation of the predictive performance, calibration, and clinical applicability of the malnutrition prediction models

The Hosmer–Lemeshow test results, with a *P*-value of 0.688, suggest no significant difference between predicted and observed malnutrition rates across subgroups, supporting the model's calibration. Internal bootstrap validation revealed a C-index of 0.793, indicating the model's effective predictive capability. The predictive accuracy of the malnutrition nomogram was further assessed through ROC analysis. For the training set (Fig. 4A) and validation set (Fig. 4B), the AUC values of the nomograms were 0.793 (95% confidence interval (CI) [0.776–0.810]) and 0.832 (95% CI [0.801–0.863]), respectively. The *P*-value of 0.564 indicates no significant difference in predictive performance between the training and validation cohorts. In contrast, the predictive model for the training set demonstrated a significantly superior AUC compared to the NRS-2002 scale, which had an AUC of 0.737 (*P* = 0.012), as illustrated in Fig. 4C. These results indicate the model's enhanced capability to predict malnutrition risk in older patients with malignant tumours. The nomogram demonstrated robust calibration within the validation cohort (Figs. 5A and 5B), indicating its excellent discriminative ability. Furthermore, bar charts were generated to assess the discriminatory power of the nomogram in identifying malnutrition

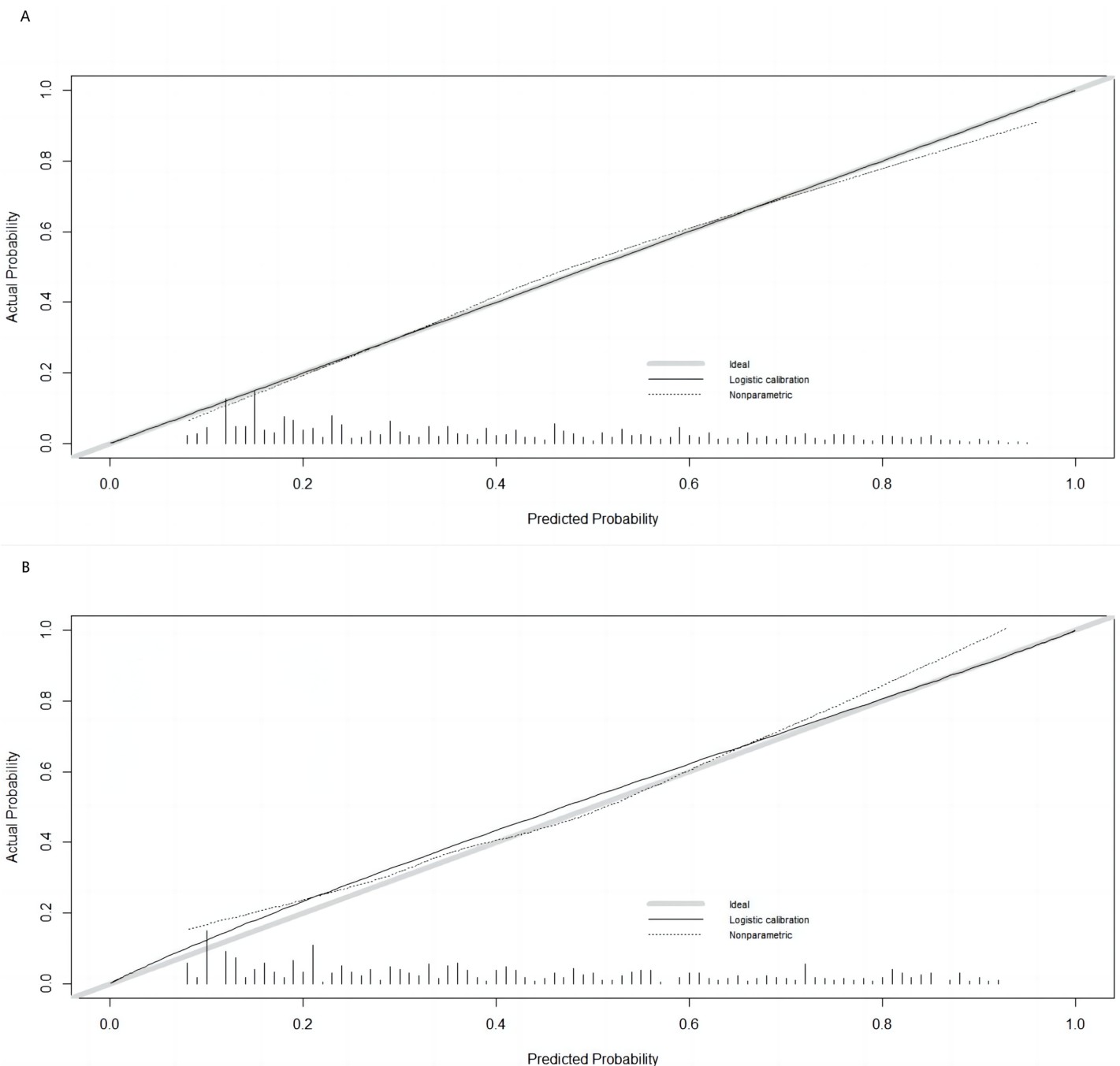

**Figure 5 Calibration curves for training and validation cohorts.** (A) Calibration curve for the training cohort; (B) calibration curve for the validation cohort. The ideal model's perfect prediction is depicted by the thich grey line, the target parameter by the black dashed line, and the model's actual performance by the solid black line. Bootstrap resampling was used (*n* = 500).

based on calculated risk scores. DCAs were conducted for the nomograms predicting malnutrition (Figs. 6A and 6B) in both the training and testing sets. Threshold probabilities ranging from 0 to 0.3 for malnutrition were identified as most advantageous for predicting malnutrition using our nomograms. The DCAs revealed that employing the

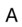

A

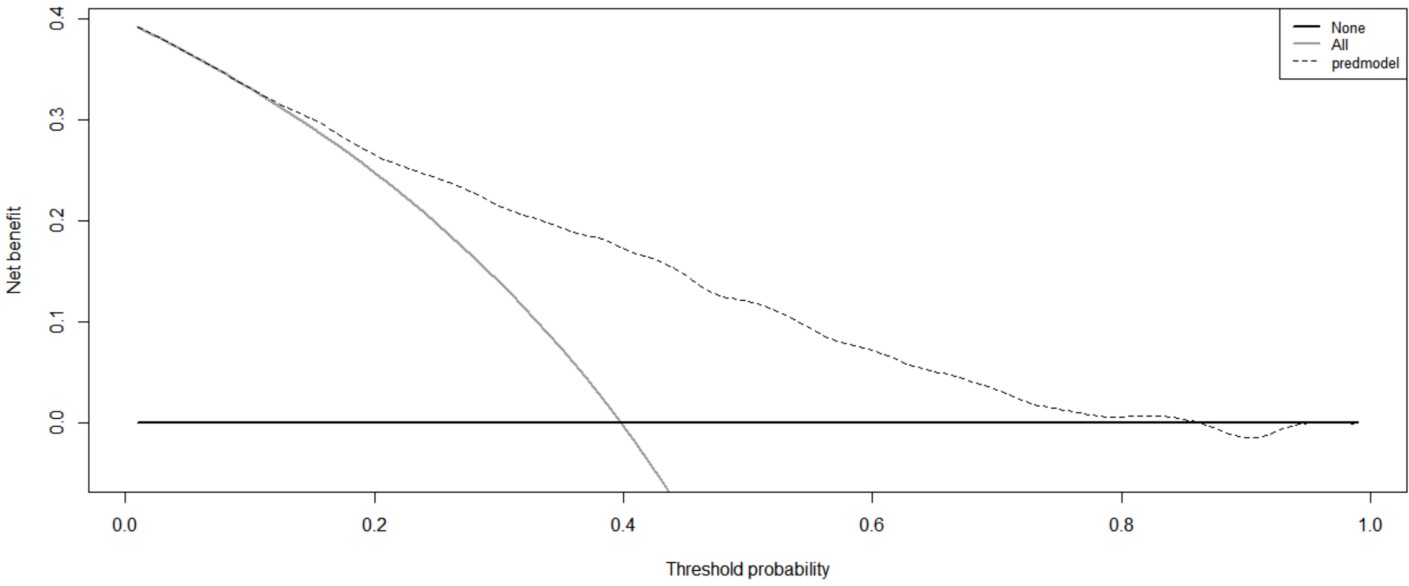

B

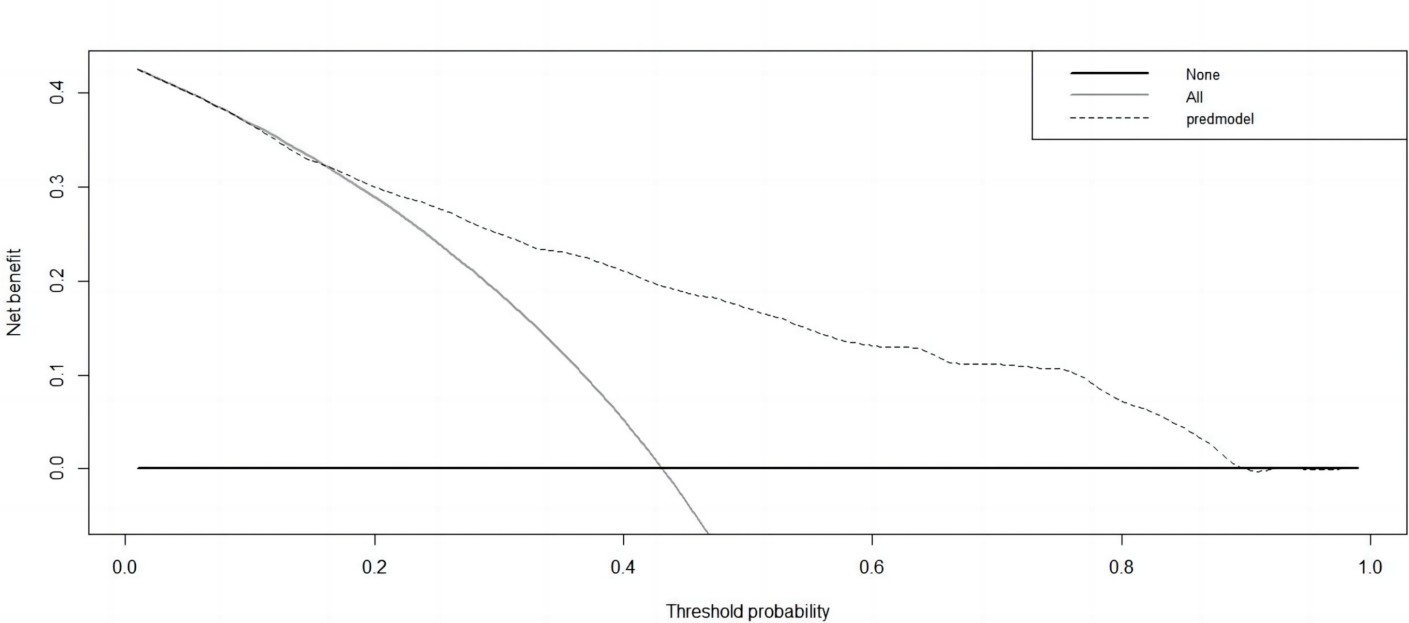

**Figure 6 Decision Curve Analysis (DCA) curve of the nomogram in the training and validation cohorts.** (A) Decision curve of the nomogram for predicting malnutrition in the training cohort; (B) decision curve for predicting malnutrition in the validation cohort. The prediction model is represented by a black dashed line; the grey solid line represents intervention in all samples, and the black solid horizontal line represents no intervention in any sample. The graph illustrates the expected net benefit per patient based on the nomogram's ability to predict malnutrition, highlighting the net benefit's increase with the model curve's extension.

nomogram for malnutrition prediction would benefit more than universally classifying patients as either malnourished or well-nourished.

## DISCUSSION

The increasing cancer-related mortality in China is largely attributed to the aging population. A significant proportion of cancer cases and deaths occur among individuals aged ≥65 years, with projections indicating that by 2030, this age group will account for nearly 70% of new cancer diagnoses (*Siwakoti et al., 2020*). Malnutrition is significantly prevalent among patients with cancer, exacerbated by both the disease and its treatment, and its probability increases with age. Approximately 66% of older patients experience malnutrition, and having cancer amplifies the risk of malnutrition by 14 times (*Van Den Broeke et al., 2018*; *Zhang & Edwards, 2019*). Despite the focus of much existing research on younger patients, there remains a notable gap in studies addressing nutritional management specifically for older cancer patients (*Presley et al., 2016*). Malnutrition significantly endangers health and increases mortality risks among older cancer patients. The incidence of malnutrition among hospitalized individuals varies between 20% and 80% (*Pavičić Žeželj et al., 2020*; *Shpata et al., 2015*). Our findings revealed that 40.42% of hospitalized patients with malignancies were malnourished, consistent with the rates reported in studies by *Arends et al. (2017)* and *Cederholm et al. (2019)*. Nevertheless, our observed prevalence is lower than the rates commonly cited in literature, where malnutrition in cancer patients, particularly those in advanced stages, has been reported to reach 6,070% (*Aapro et al., 2014*; *Silva et al., 2015*). This disparity may be attributed to variability in our inclusion criteria, particularly the exclusion of older patients with severe health conditions. Our findings align with studies using the NRS-2002 to assess malnutrition risk, showing high prevalence among older cancer patients, influenced by age, Hukou status, gender, and tumour type. Specifically, malnutrition rates increase with age due to comorbidities and age-related impacts on nutrient intake and absorption (*Okayama et al., 2022*). Additionally, malnutrition and obesity are more common in peri-urban and rural areas (*Hernández-Vásquez et al., 2022*; *Chadare et al., 2019*). Our investigation revealed that malnutrition is more common among rural older individuals, influenced by economic constraints, dietary habits, limited nutritional knowledge, lifestyle challenges, and restricted healthcare access (*Mumme et al., 2019*; *Kushwaha et al., 2020*). While literature suggests these factors contribute to nutritional issues, specific data supporting their direct impact in this context are limited.

Among different tumour types, those with gastrointestinal cancers show the highest malnutrition rates, likely due to the critical role of the gastrointestinal system in nutrient absorption and metabolism (*Pavičić Žeželj et al., 2020*). This study involved 3,387 patients aged ≥65 with cancer, all in overall good health. We analyzed 26 variables and used the GLIM criteria for diagnosing malnutrition, based on comprehensive patient data collected at a major tertiary hospital in Southwest China. We developed and validated a novel clinical prediction model for malnutrition across various periods and independent cohorts, demonstrating significant potential for widespread clinical application.

Multivariate logistic regression analysis identified several significant risk factors for malnutrition, including age, TNM stage, presence of digestive system tumours, KPS, alcohol consumption, combined thoracic and abdominal fluid effusion, HGB and Cr levels, and the NLR. Tumour stages were grouped into stages I–III (non-metastatic) and stage IV (metastatic) to address uneven distribution and enhance the study's validity. Higher tumour stages, indicative of increased malignancy (*Ravasco et al., 2004*), are associated with a higher risk of symptoms such as cancer pain, anorexia, nausea, and vomiting, along with severe complications like gastrointestinal disorders. These factors contribute to inadequate nutritional intake and absorption in patients with advanced cancer, often resulting in multiple prolonged hospitalizations. Our findings confirm the link between tumour severity and malnutrition risk, consistent with existing literature (*Dai et al., 2023*; *Wang et al., 2023*).

Patients with digestive system tumours experience significant impairments in appetite and nutrient absorption compared to those with tumours in other systems, primarily due to complex factors affecting nutritional intake, leading to higher malnutrition rates among those with digestive cancers (*Wang et al., 2023*; *Kaźmierczak-Siedlecka et al., 2023*). While existing research has extensively focused on malnutrition in patients with digestive system tumours, less attention has been given to other tumour locations. Our study addressed this gap by examining malnutrition across all older patients with malignant solid tumours, revealing distinct prevalence between digestive and non-digestive tumours. However, the physiological, pathological, and clinical differences between tumour types, along with other influential factors, may introduce biases into our results, representing a limitation of our study. We also identified four key predictors of malnutrition—KPS score, alcohol consumption, presence of pleural and abdominal fluids, and HGB levels—all closely associated with malnutrition prognosis in older cancer patients.

Clinically, the KPS score serves as a valuable indicator of functional capacity in cancer patients, where higher scores correlate with improved survival outcomes, enhanced quality of life, and reduced malnutrition risk (*Sun, Shen & Wang, 2023*; *Ruiz-Sánchez et al., 2012*). Alcohol consumption is recognized as a lifestyle factor linked to increased mortality in various conditions, including cancers and liver cirrhosis (*Ferrari et al., 2014*). Co-occurring diseases among frequent alcohol consumers often result in malnutrition due to compromised nutrient absorption. Additionally, malnutrition has been associated with complications like ascites and exacerbated liver dysfunction (*García-Rodríguez et al., 2018*; *McSharry et al., 2021*; *Kathrani, Sánchez-Vizcaíno & Hall, 2019*). This study indicated that in patients with combined thoracic and abdominal fluid effusion, tumour-related factors significantly impact malnutrition compared to those related to the patients themselves.

HGB is recognized as a biomarker for malnutrition in older individuals (*Zhang et al., 2017*). In our analysis, we observed a significant relationship between anaemia and malnutrition, highlighting the importance of monitoring HGB levels as a critical factor in assessing the nutritional status of patients with malignant tumours. A systematic review further validated HGB as a valuable biomarker for malnutrition in older adults, linking lower HGB levels to decreased mobility. Decreased HGB levels are widely recognized as indicators of inflammation and malnutrition, a correlation supported by studies from *Lu*

*et al. (2019)*, *Fernández-Ruiz et al. (2020)*. These findings underscore the need for increased clinical awareness of anemia in cancer patients.

Systemic inflammation can be quantified using blood cell markers, providing a cost-effective means of evaluation. Recent research has explored inflammatory biomarkers for predicting survival outcomes in cancer patients, assessing nutritional risks, and guiding treatment strategies (*Schiefer et al., 2022*). Our study included serum creatinine levels as indicators of renal function. However, literature on the coexistence of renal insufficiency and malnutrition in cancer patients is limited, with few studies addressing both populations (*Kimaro et al., 2019*; *Otaki et al., 2022*; *Yelken et al., 2010*). Further investigation is needed to clarify the relationship between renal insufficiency severity and malnutrition incidence.

Consistent with previous research, our analysis highlighted the NLR as a marker of systemic inflammation, demonstrating its predictive value for malnutrition among older cancer patients. Our findings indicated that NLR significantly predicts malnutrition in this demographic, with heightened inflammatory responses in cancer patients increasing nutrient consumption and infection rates, thereby exacerbating malnutrition risk (*Sun et al., 2016*). Moreover, neutrophils influence tumor growth and metastasis through cytokine secretion, contributing to malnutrition development. The decline in lymphocyte count compromises immune function and heightens the risk of tumor metastasis, further depleting nutrients and disrupting digestive functions (*Dai et al., 2023*; *Jia et al., 2021*; *Shigeto et al., 2020*).

Despite the rigorous design of our study and comprehensive conclusions, significant variables identified in univariate analysis, such as gender and marital status, were not reflected in multivariate results. This inconsistency may arise from omitting significant confounding covariates, leading to residual confounding and influencing estimated effects. Additionally, the lack of intricate interactions between predictors may obscure the full extent of how variables influence each other. A comprehensive validation of the regression model's robustness was not conducted, which is crucial for verifying adherence to ordinary least squares regression assumptions. These limitations could contribute to the disparities observed between univariate and multivariate analyses. Subsequent research should focus on enhancing model design by increasing sample size and refining variable selection to achieve more reliable outcomes.

Based on evaluating 10 specified variables, our study introduces a concise, efficient column-line graph designed to assess malnutrition risk among individuals aged ≥65 with solid malignant tumours. This novel approach appears unique in the existing literature, offering a precise tool for clinicians to evaluate malnutrition risk in this demographic. Our experimental design stratified participants into training and validation sets with comparable baseline characteristics. The models' discriminative performance was assessed using ROC curve analysis, yielding an AUC of 0.792 for the development set and 0.832 for the validation set. These findings indicate enhanced overall discriminative capacity. Compared to the standard NRS-2002 scale, our model demonstrated higher predictive accuracy with an AUC of 0.732, supported by calibration curves and the

Hosmer–Lemeshow goodness-of-fit test ($P > 0.05$). Consequently, the proposed nomogram offers significant predictive value for clinical practice.

Despite its extensive sample size and broad variable consideration over a prolonged period, this comprehensive retrospective study carries inherent limitations typical of retrospective analyses. Its concentration on older patients with solid tumours aged ≥65 at a single tertiary care hospital in Southwest China may restrict the generalizability of our findings to a broader demographic. Moreover, the predictive model requires additional validation through an extensive multicentre, prospective cohort study. Furthermore, the selection criteria for inclusion lacked comprehensiveness, notably omitting factors such as patients' socioeconomic status, dietary patterns and habits, antitumour treatments, serum prealbumin levels, and cardiac function classification (CFG). Consequently, a large-scale multicentre prospective study will incorporate these variables, aiming to enhance the model's clinical generalizability.

## CONCLUSIONS

This study utilized multiple regression analysis to identify the risk factors for malnutrition in older patients aged ≥65 with malignant tumours. A clinical prediction model was developed, calibrated, and validated that exhibited predictive solid accuracy and clinical applicability, facilitating the implementation of effective early intervention strategies for malnutrition prevention.

### Funding
The authors received no funding for this work.

### Competing Interests
The authors declare that they have no competing interests.

### Author Contributions
- Xuexing Wang conceived and designed the experiments, performed the experiments, analyzed the data, prepared figures and/or tables, and approved the final draft.
- Jie Chu performed the experiments, prepared figures and/or tables, and approved the final draft.
- Chunmei Wei performed the experiments, prepared figures and/or tables, authored or reviewed drafts of the article, and approved the final draft.
- Jinsong Xu performed the experiments, prepared figures and/or tables, and approved the final draft.
- Yuan He performed the experiments, prepared figures and/or tables, and approved the final draft.
- Chunmei Chen performed the experiments, prepared figures and/or tables, and approved the final draft.

## Human Ethics

The following information was supplied relating to ethical approvals (*i.e.*, approving body and any reference numbers):

This study was carried out with the oversight and endorsement of the First People's Hospital of Anning, which is affiliated with Kunming University of Science and Technology (Ethical Approval No. 2024-020 (Self-horizontal)-01)).

## Data Availability

Raw data can be found in the Supplemental Material.

## Supplemental Information

Supplemental information for this article can be found online at http://dx.doi.org/10.7717/peerj.18685#supplemental-information.

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
