# Peer review of "Construction and validation of a predictive model for the risk of malnutrition in hospitalized patients over 65 years of age with malignant tumours: a single-centre retrospective cross-sectional study"

_PeerJ, doi:10.7717/peerj.18685_

## Round 0.1 · original submission · Major Revisions

The authors are requested to carefully revise the manuscript and answer the questions raised by the reviewers.

Reviewer 1 ·

Basic reporting

1. Basic Reporting
Line 54: This definition requires referencing
Line 60: ‘especially the older ones’ – rephrase with more appropriate English e.g. especially those of older age.
Line 69: Despite its 70 significance, universally accepted diagnostic criteria for malnutrition are yet to be 71 established.
I disagree with this statement.
Risk tools for older adults = MST for older adults >65yo
There are many accepted and validated tools that are used in the diagnosis of malnutrition e.g SGA, PG-SGA (cancer specific), ICD-10 Codes, SNAQ, MNA and MNA-SF to name a few.
The PG-SGA or even the PG-SGA short form can be used in older populations with cancer appropriately.
https://bmcgeriatr.biomedcentral.com/articles/10.1186/s12877-021-02662-4
https://pubmed.ncbi.nlm.nih.gov/34893024/

Experimental design

2. Experimental design
Methods
Is the use of NRS and GLIM standard of care for patients admitted to the hospital? If not, how was this data collected as it was retrospective? The design of the research is quite unclear and confusing.
Line 95: 3387 consecutive older inpatients
This should go in the results.
Line 117: After applying these criteria, 3387 patients were included in the study, as depicted in the flow chart (Figure 1).
This is repetitive and should go in the results.
Line 120: The calculation of an adequate sample size for predictive studies, encompassing both modelling and validation phases, depends on the number of outcome events. Following guidelines from predictive modelling research
These need to be referenced. Which predictive models are you using?
Line 155: low BMI (Asian standard: BMI
Provide the reference for the Asian standard BMI cut-offs.
Line 157: Decreased muscle mass (below the standard values for different body composition measures)
How was this measured? Was is subjective physical assessment or were there objective measures/tools used?
Line 161: disease/inflammation burden (associated with either acute disease or trauma, or chronic conditions).
What cut off score did you use to determine inflammation?
Line 172: Column 173 plots were generated using the "rms" package in R software (version R4.2.0).
Reference the software that you have used.
Line 174: SPSS
What version of SPSS? Please reference.
Line 180: inspection and the Hosmer-Lemeshow test.
Reference and describe the Hosmer-Lemeshow test – many people won’t know what this is.

Validity of the findings

3. Validity of the findings
Results
The first paragraph of your results should describe the characteristics of your population in the first instance and then be presented as a demographics table e.g. age, gender, cancer type, cancer TNM. Bring up Table 1 to the first paragraph to describe your sample.
The pie graphs in Figure 2 need more comprehensive titles that better explain what is being demonstrated e.g. 2d. The percentage of patients with malnutrition based on cancer type (body system)
Line 219: differences were observed in drinking history
Are you referring to alcohol consumption? Please correct this.
Line 240: Supplementary Figure 1.
Why does this appear in a different size font?
Discussion
Line 293: approximately 66% of older patients with malignant tumours experience malnutrition, and having cancer amplifies the risk of malnutrition by 14 times
Just be cautious of this sentence as it you have said malignancy/cancer twice. I think this should be reworded for better clarity.
Line 301: indicated that 40.42% of hospitalized patients were malnourished
Is your population only those with a malignancy? Then state this as this would overinflate the malnutrition prevalence of the hospitalised patient cohort.
Line 301: aligns 302 with some contemporary studies but is less than the rates frequently cited.
What studies? There are none referenced here.
Line 305: Previous research indicates the prevalence of malnutrition in hospitalized patients 306 ranges from 20% to 80%. Our results of a 40.42% malnutrition rate among 307 admitted patients resonate with some current studies
Remove, this is repetitive as per Line 299 to 301.
Line 321: This disparity can be attributed to various factors
This needs referencing otherwise how do you know this?
Line 472: Consequently, our predictive is positioned
Please reword appropriately.

Additional comments

I believe that there are many robust evidence-based and validated tools to determine both malnutrition risk and also diagnose malnutrition in adult populations with cancers, including those who are older. I need to be convinced that there is a benefit in recreating yet another tool that seems complex when tools already exist that can be used effectively.

The discussion needs to be rewritten. It is repetitive and not clear or summarised.

Reviewer 2 ·

Basic reporting

No comment

Experimental design

Please clarify the tools used in this study to identify malnutrition, this is an important information. In the abstract, the authors stated that the ESPEN 2015 criteria of malnutrition were used. However, in the method section, they declared that GLIM criteria were used. Please note that ESPEN 2015 and GLIM were two different criteria. The author also should give detailed information about the etiology and phenotypic factors, such as how many patients had weight loss, how many had low BMI. Low muscle mass is an important phenotypic factor in the GLIM criteria. How did the authors evaluate the muscle mass? I speculated that in a retrospective study, many patients would be exclude because of the lack of data of muscle mass. How did the authors deal with this problem.

Validity of the findings

The authors should be careful in presenting the results, the prevalence of malnutrition was 38.4% in the abstract, however, the authors claimed that it is 40.49% in the results section. It is contradictory. Besides, 1369/3387 equals 40.42%, not 40.49%.

Additional comments

No

---

## Round 0.2 · Major Revisions

The authors are requested to further carefully revise the manuscript and answer the questions raised by the reviewers.

Reviewer 1 ·

Basic reporting

Please reference the standard reference values you used in the revised section on muscle mass body composition techniques.

Similarly, the standard inflammation levels need referencing too.

The reference for Hosmer and Lemeshow should be this: Hosmer, David W.; Lemeshow, Stanley; Sturdivant, Rodney X. (2013). Applied logistic regression. Wiley series in probability and statistics (3 ed.). Hoboken, NJ: Wiley. ISBN 978-0-470-58247-3.

In the discussion, the use of the word malnourishment should be changed to malnutrition.

Patients with tumours of the digestive system experience more significant impairments in appetite, digestion, and absorption nutrient absorption compared to those with tumours in other systems - repetitive, this has already been said in a paragraph before.

I think your discussion is still very repetitive and requires further revision. It is also still very long. You can cull a lot of what is written.

Experimental design

Improved documentation of experimental design

Validity of the findings

Improved documentation

Additional comments

N/A

·

Basic reporting

This article used over 3000 patients with malignant tumors to develop and validate a clinical model for malnutrition prediction, get fairly good results.

Experimental design

1. How were the training and validation groups assigned, randomly or by some ways, with or without human intervention? As we know, training set usually perform better than test set, but the results in this article reversed. So there may be reason to doubt that the allocation of training and validation groups in this article makes sense.
2. Is that reasonable for mixing malignant patients from multiple systems in instrumental analyses?
3. KPS score seems not mentioned in the methods part, but appeared in the results part, which may cause confusion.

Validity of the findings

1.Figure2 seems to appear a bit redundant, since the information has showed in Table1-2.
2.In table 1, it seems better to add the information of the ending variable, malnutrition or not.
3.In table 2, the first line of "Nourished or Malnourished" need to unify the terminology with the malnutrion in the text.

Additional comments

Regarding the clinical relevance of this article, the nutritional status of elderly patients may be known by scoring or by haematology, so why is there a need for multi-parameter modelling, and does it really bring convenience to the clinic?

Reviewer 4 ·

Basic reporting

No comment.

Experimental design

1. Line 111: “This retrospective cohort study…”
The title: “Construction and validation of a predictive model for the risk…: a single-centre retrospective case-control study”.

Cohort or case-control study design? Pls confirm and revise them. Actually, the data sources of the diagnostic model are mainly cross-sectional studies.

Validity of the findings

No comment.

Additional comments

1. “Novelty”, Limited novelty exists in the current study. Similar publications could be retrieved in PubMed.
[1] Duan R, Li Q, Yuan QX, Hu J, Feng T, Ren T. Predictive model for assessing malnutrition in elderly hospitalized cancer patients: A machine learning approach. Geriatr Nurs. 2024 Jul-Aug;58:388-398. doi: 10.1016/j.gerinurse.2024.06.012.
[2] Dai T, Wu D, Tang J, Liu Z, Zhang M. Construction and validation of a predictive model for the risk of three-month-postoperative malnutrition in patients with gastric cancer: a retrospective case-control study. J Gastrointest Oncol. 2023 Feb 28;14(1):128-145. doi: 10.21037/jgo-22-1307.

Annotated reviews are not available for download in order to protect the identity of reviewers who chose to remain anonymous.

---

## Round 0.3 · Minor Revisions

The authors are requested to carefully revise the manuscript and answer the final questions raised by the reviewers.

Reviewer 1 ·

Basic reporting

The authors have now added in ISAK low muscle mass determinants - were staff trained in the ISAK methods with certification? I don't think you can reference this if they weren't adequately trained. There are other cut-offs documented extensively through the literature.

Also they have added a reference for standard inflammatory cut-offs in diabetes patients, I would urge the authors to source a reference in the cancer cohort that they are developing a tool about.

Experimental design

Improved adequately.

Validity of the findings

Improved adequately.

Additional comments

Improved adequately.

·

Basic reporting

no comments

Experimental design

I am grateful for your response. I believe it is acceptable to differentiate between the training and validation groups based on the timeline. However, I recommend including this information in the article itself. I did not observe it in the abstract or methodology section.

Validity of the findings

no comments

Additional comments

no comments

Reviewer 4 ·

Basic reporting

No comment.

Experimental design

No comment.

Validity of the findings

No comment.

Additional comments

All my concerns were addressed satisfactorily by the authors, thanks. Compared to the STROBE checklist for cohort study, I would like recommend STROBE checklist for cross-sectional study, which may be more suitable for the current study design.

---

## Round 0.4 · accepted · Accept

After revisions, two reviewers agreed to publish the manuscript. There is one reviewer left with a minor revision, and I think the author has responded adequately. I also reviewed the manuscript and found no obvious risks to publication. Therefore, I also approved the publication of this manuscript.

·

Basic reporting

no comment

Experimental design

no comment

Validity of the findings

no comment

Additional comments

The author has revised it as suggested and I think it basically meets the requirements for publication.